# Development of a Prototype Lateral Flow Immunoassay for Rapid Detection of Staphylococcal Protein A in Positive Blood Culture Samples

**DOI:** 10.3390/diagnostics10100794

**Published:** 2020-10-07

**Authors:** Arpasiri Srisrattakarn, Patcharaporn Tippayawat, Aroonwadee Chanawong, Ratree Tavichakorntrakool, Jureerut Daduang, Lumyai Wonglakorn, Aroonlug Lulitanond

**Affiliations:** 1Centre for Research and Development of Medical Diagnostic Laboratories, Faculty of Associated Medical Sciences, Khon Kaen University, Khon Kaen 40002, Thailand; arpasr@kkumail.com (A.S.); patchatip@kku.ac.th (P.T.); aroonwad@kku.ac.th (A.C.); ratree.t@kku.ac.th (R.T.); jurpoo@kku.ac.th (J.D.); 2Clinical Microbiology Unit, Srinagarind Hospital, Khon Kaen University, Khon Kaen 40002, Thailand; wlumya@kku.ac.th

**Keywords:** bloodstream infection, lateral flow immunoassay, positive blood culture samples, rapid detection, *Staphylococcus aureus*

## Abstract

Bloodstream infection (BSI) is a major cause of mortality in hospitalized patients worldwide. *Staphylococcus aureus* is one of the most common pathogens found in BSI. The conventional workflow is time consuming. Therefore, we developed a lateral flow immunoassay (LFIA) for rapid detection of *S. aureus*-protein A in positive blood culture samples. A total of 90 clinical isolates including 58 *S. aureus* and 32 non-*S. aureus* were spiked in simulated blood samples. The antigens were extracted by a simple boiling method and diluted before being tested using the developed LFIA strips. The results were readable by naked eye within 15 min. The sensitivity of the developed LFIA was 87.9% (51/58) and the specificity was 93.8% (30/32). When bacterial colonies were used in the test, the LFIA provided higher sensitivity and specificity (94.8% and 100%, respectively). The detection limit of the LFIA was 10^7^ CFU/mL. Initial evaluation of the LFIA in 20 positive blood culture bottles from hospitals showed 95% agreement with the routine methods. The LFIA is a rapid, simple and highly sensitive method. No sophisticated equipment is required. It has potential for routine detection particularly in low resource settings, contributing an early diagnosis that facilitates effective treatment and reduces disease progression.

## 1. Introduction

*Staphylococcus aureus* is one of the most important bacterial pathogens, causing a variety of diseases such as food poisoning, pneumonia, wound and bloodstream infections. It is the Gram-positive pathogen most frequently recovered from positive blood cultures [1]. *Staphylococcus aureus* bacteremia (SAB) is associated with significant morbidity and mortality, especially in patients of intensive care units. The global incidence rate of SAB infection ranges from approximately 10 to 65 cases/100,000 population per year with a mortality rate of 22% to 48% [2]. Delays in the identification of the organism lead to inappropriate therapeutic options, progressive stages of severity and decreased survival rate. Therefore, rapid species identification for early diagnosis is important to facilitate effective treatment and reduce the severity of the disease. 

The standard method for diagnosis of *S. aureus* requires culturing on an agar plate and biochemical tests. This conventional workflow is time consuming, taking a few days. Rapid methods for bacterial identification, including Matrix-Assisted Laser Desorption/Ionization-Time of Flight mass spectrometry or other automated methods, such as the VITEK 2 MS system (bioMérieux, Marcy l’Etoile, France), MicroScan Walkaway system (Siemens Healthcare Diagnostics, Sacramento, CA, USA) or BDPhoenix^™^ (Becton Dickinson, Sparks, MD, USA), have been increasingly used in the routine diagnostic laboratory [3,4,5]. However, these systems are very expensive and are not widely available, especially in low-resource settings. Several molecular techniques, including PCR-based methods, were tested in positive blood culture samples with high sensitivity and specificity within 3 h [6,7]. However, they also require special equipment and well-trained operating personnel. The latex agglutination test is rapid and widely used for *S. aureus* detection but it can only be performed on pure cultures or colonies. It also requires prior isolation of the pathogen [8,9]. Therefore, rapid and simple methods are still needed to confirm identification of *S. aureus.*

Surface protein A was the first surface protein identified in *S. aureus* due to its ability to bind with immunoglobulins (Igs) [10]. The five N-terminal repeat domains of protein A bind with the Fc portion of immunoglobulins. This protein is a constituent in the cell wall of *S. aureus*. About 90% of protein A is found in the cell wall and the remaining 10% is free in the cytoplasm of the bacteria [11]. It was found in 90% to 100% of *S. aureus* strains [12,13]. Therefore, protein A is used as a target for detection and identification of *S. aureus* [13]. 

Chemical and biological sensing are important tools for diagnostics in medical sciences. Plasmonic nanoparticles, such as gold nanoparticles, are generally reported to be biocompatible, which can be available for visual detection. The use of a paper-based format has several advantages, including easy fabrication, optic transparency, biocompatibility and being lightweight and disposable technology. Therefore, the combination of plasmonic nanoparticles and paper-based leads to simple, single use and cost efficient analytical devices, which is useful to develop the point-of-care devices further [14,15]. 

A lateral flow immunoassay (LFIA) is a diagnostic device based on immunological reaction (antigen–antibody interaction) and chromatography (capillary action) of a labeled analyte (gold nanoparticles) through multiple membranes, including sample, conjugate, detection and absorbent pads [16,17]. This method is a low-cost, simple, rapid and portable detection device popular in several fields, such as agriculture, food, environmental sciences and biomedicine [18]. A variety of biological samples can be tested by LFIA methods, such as urine, saliva, sweat, serum, plasma, other fluids and whole blood [17]. In addition, it can be used for point-of-care testing by general staff and no special equipment is required. The LFIA technique has also been used to detect protein A of *S. aureus* [19,20]. However, it has not yet been used for fully clinical specimens, especially the positive blood culture bottles. We therefore developed an in-house LFIA using protein A as the target for rapid identification of *S. aureus* from both colonies and simulated blood cultures. We have done a preliminary evaluation of the performance of the test in a hospital setting. This rapid, reliable, and accurate diagnostic test will lead to appropriate antibiotic treatment, increased survival rates and reduced duration of hospitalization.

## 2. Materials and Methods

### 2.1. Bacterial Strains

A total of 90 bacterial isolates were tested, including 58 *S. aureus* and 32 non-*S. aureus* isolates from Srinagarind Hospital, Khon Kaen University, Thailand between 2010 and 2019 (Table 1). All clinical isolates were identified by conventional biochemical tests and PCR methods and kept in skimmed milk with 15% glycerol at −20 °C until used. Reference strains of *S. aureus* (NCTC10442) and *S. haemolyticus* (CNSP40) were used as protein A-positive and -negative controls, respectively.

This study was approved (26 February 2019) by the Ethics Committee of Khon Kaen University (project number HE611605).

### 2.2. Establishment and Assembly of the LFIA

#### 2.2.1. Preparation of Gold Nanoparticles

A colloidal of 13 nm gold nanoparticles was prepared by using a citrate reduction method according to a previous report [21] (chloroauric acid (HAuCl_4_) was purchased from Sigma-Aldrich (St. Louis, MO, USA)). The colloidal gold nanoparticle was filtered through a 0.45 µm membrane filter (Whatman, GE Healthcare UK Ltd., Little Chalfont, UK) after it was allowed to cool to room temperature. The size of the gold nanoparticles was determined using transmission electron microscopy (FEI Tecnai G2 20, Hillsboro, OL, USA) and measured (100 particles) by ImageJ software (version 1.48v) (National Institute of Health, Bethesda, MD, USA). In addition, the UV–Vis absorption measurement was carried out in the range from 400 to 800 nm.

#### 2.2.2. Optimization of Gold Nanoparticle Conjugation

The anti-protein A polyclonal antibody (pAb) (Arista Biological, Allentown, PA, USA) was diluted to 0, 20, 40, 60, 80, 100, 120, 140, 160, 180 and 200 µg/mL in a 10 mM phosphate buffer with a pH of 7.4, then 50 μL of colloidal gold nanoparticles (pH 8.0) were added to 5 μL of each antibody dilution and kept at room temperature for 10 min. Next, 10 μL of 10% NaCl (RCI Labscan Limited, Mueng, Samutsakorn, Thailand) was added to each vial and the color changes were observed. Incompletely coated colloidal gold fallout and the solution turned black or blue, while completely coated particles remained stable and the solution remained red in color. The lowest concentration of antibody giving no color change was considered to be the optimal concentration for stabilizing the colloidal gold nanoparticle.

#### 2.2.3. Preparation of Gold Nanoparticle Conjugate Antibody 

The gold nanoparticles were coated with anti-protein A pAb (Arista Biological, Allentown, PA, USA) following Kestrel Bio Sciences (Pathumthani, Thailand) Co., Ltd. protocol with slight modification [22]. Briefly, 1 mL of gold nanoparticles was added to 100 μL of 180 µg/mL of anti-protein A pAb solution, mixed for 5 s and placed at room temperature for 30 min. After that, the coated gold nanoparticles were blocked by adding 10% bovine serum albumin (BSA) (Sigma-Aldrich, St. Louis, MO, USA) to a final concentration of 1% (*w/v*) and placed at room temperature for 10 min. The conjugate solution was centrifuged at 14,000× *g*, 4 °C for 20 min to remove free antibodies, and the pellet was resuspended in 0.5 mL of 10 mM phosphate buffer (PB) with a pH of 7.4 containing 1% BSA, 20% sucrose (Merck, Darmstadt, Germany) and 0.05% sodium azide (Fluka, Dorset, UK). The extent of conjugation between gold nanoparticles and antibodies was evaluated by a UV–Vis spectrophotometer (Eppendorf BioSpectrometers, Hamburg, Germany). Twenty µL of labeled conjugate was dropped on a piece of glass microfiber filter conjugate pad with a size of 0.4 × 0.9 cm (GF33; Whatman Schleicher & Schuell, Dassel, Germany), dried at 37 °C for 2 h and stored at 4 °C for further uses.

#### 2.2.4. Preparation of LFIA Strips

The LFIA strip is composed of four parts: sample pad, nitrocellulose membrane (NCM; detection pad), conjugate pad and absorbent pad [15]. The NCMs and sample pads tested for the efficacy included UniSart^®^ CN95 (pore size of 15 μm with a flow rate of 95 s/4 cm) and UniSart^®^ CN140 NCMs (pore size of 8 μm with a flow rate of 140 s/4 cm) (Sartorius Stedim Biotech SA, Goettingen, Germany); Cytosep 1660 (Pall Gelman Sciences, Champs-sur-Marne, France) and Milipore C048 sample pads (Millipore, MA, USA). In addition, five different running buffers were tested: buffer 1 (50 mM Tris-HCl (Bio Basic Inc., Markham ON, Canada), 200 mM NaCl and 1% Triton X-100 (Panreac Química SA, Barcelona, Spain)); buffer 2 (50 mM Tris-HCl and 1% Triton X-100); buffer 3 (50 mM Tris-HCl and 1% Tween 20 (Merck, Darmstadt, Germany)); buffer 4 (5 mM phosphate buffer with a pH of 7.4 and 1% Triton X-100) and buffer 5 (5 mM phosphate buffer with a pH of 7.4 and 1% Tween 20). Before used, the sample pads were soaked in sample pads treatment buffer (0.25 M tris-aminomethane (Merck, Darmstadt, Germany), 0.1% BSA, 0.1% Tween 20 and 0.1% sodium azide) for 1 min and dried at 37 °C for 3 h.

After testing with different pad materials and buffers, anti-protein A pAb (1.0 mg/mL) and goat anti-chicken antibody (1.0 mg/mL) (Lampire Biological Laboratories, Pipersville, PA, USA) were dispensed onto the NCM detection pad (UniSart^®^ CN140) test and control lines, respectively, using a lateral flow dispenser (XYZ3000 Dispensing Platform; BioDot Inc., Irvine, CA, USA) (1 µL/cm) and dried at 37 °C for 1 h. The treated glass fiber conjugate pad (GF33), Cytosep 1660 sample pad, UniSart^®^ CN140 NCM and Whatman ABS PAD #470 absorbent pad were assembled onto the backing card with a 0.2 cm overlap between successive components. Then, the LFIA strip was cut into narrower 4 mm × 6 cm strips using a CM5000 Guillotine Cutter (BioDot Inc., Irvine, CA, USA).

### 2.3. PCR-Based Identification of S. aureus by Amplification of the nuc Gene

Bacterial DNA was extracted using achromopeptidase enzyme (Sigma-Aldrich, St. Louis, MO, USA) according to the method of Shittu et al. [23]. After a 1 min centrifugation, the supernatant was used as the DNA template in the PCR reaction. The amplification of the *nuc* gene was carried out using a primer set of *nuc* forward 5′-TTAAGTGCTGGCATATGTATGGCAATCGTTTC-3′ and *nuc* reverse 5′-CACCATCAATCGCTTTAATTAA TGTCGCAGGTTC-3′ (this study). The PCR reaction was carried out in a total volume of 25 µL comprised of 0.5 µM of each primer (forward and reverse) (synthesized by Pacific Science, Bangkok, Thailand), 1× PCR buffer (Vivantis Technologies, Subang Jaya, Malaysia), 0.2 mM dNTPs (Vivantis Technologies, Subang Jaya, Malaysia), 2 mM MgCl_2_ (Vivantis Technologies, Subang Jaya, Malaysia), 1 unit of *Taq* DNA polymerase (Vivantis Technologies, Subang Jaya, Malaysia) and 2 µL of DNA template. Cycling conditions were 95 °C for 5 min, followed by 30 cycles of 95 °C for 1 min, 55 °C for 1 min, 72 °C for 1 min and an additional 10 min at 72 °C.

### 2.4. Detection Limit of the LFIA

The bacterial suspension, grown in sterile culture fluid from Render pre-incubated aerobic culture bottles (Zhuhai Meihua Medical Technology Limited, Zhuhai, China) to a turbidity of 0.5 McFarland standard, was serially 10-fold diluted from 10^0^ to 10^−4^ and bacterial colony counts were performed after plating to Mueller–Hinton agar. One hundred µL of each dilution was pipetted onto a sample pad of the LFIA strip to determine the detection limit of the test. After 15 min, the signal of the strip was noted visually. In addition, 10 mM phosphate buffer, pH 7.4, spiked with various concentrations of purified protein A (595.24, 297.62, 148.81, 72.52, 37.14, 7.38, 0.738, 0.074 nM) (Arista Biological, Allentown, PA, USA) were also tested with the LFIA strips. The test line and/or control line would appear as a pink-purple color. The presence of both control and test lines indicated a positive result, whereas that of only the control line indicated a negative result.

### 2.5. Detection of S. aureus Directly from Bacterial Colonies and Spiked Blood Culture Samples

Bacterial colonies and blood culture samples spiked with the 58 and 32 isolates of *S. aureus* and non-*S. aureus*, respectively (Table 1), were used to investigate the efficacy (sensitivity and specificity) of the LFIA strips for direct detection of *S. aureus*.

#### 2.5.1. Testing with Bacterial Colonies

A 10 µL loop full of bacterial colonies cultured on blood agar (Oxoid, Hampshire, UK) at 35 to 37 °C for 18 to 24 h was suspended in 1.5 mL microcentrifuge tubes containing 100 µL of running buffer. The suspension was mixed to homogeneity before being applied to the sample well of an LFIA strip. The result was read within 15 min at room temperature.

#### 2.5.2. Testing with Spiked Blood Culture Samples

Overnight bacterial colonies cultured on blood agar were suspended in 0.85% NaCl solution to 0.5 McFarland turbidity and diluted to 1:1000 with 0.85% NaCl solution. Subsequently, 5 μL of the bacterial solution was mixed with 200 μL of human blood from healthy volunteers, inoculated into 750 μL of sterile culture fluid from Render pre-incubated aerobic culture bottles (final inoculum of ~800 CFU/mL) and incubated at 37 °C for 24 h. Bacterial antigen extraction was performed by a simple boiling method: 50 µL of the sample was boiled for 3 min in a heat block at 100 °C, allowed to cool for 5 min, then 50 µL of diluent (0.3 M arginine with a pH of 4.5 (Lobachemie, Mumbai, India), 0.3 M glycine with a pH of 4.5 (Merck, Darmstadt, Germany) and 0.03 M citrate with a pH of 3.5 (BDH Chemicals Ltd., Poole, UK)) added for protein A-IgG complex dissociation. One hundred μL of running buffer was mixed into the tube before the sample was tested using the LFIA strip. The strip was dipped into the mixture and the result was read visually within 15 min (Figure 1). The LFIA results were compared with conventional biochemical and PCR tests. The PCR was used as the reference method for the presence of *S. aureus.*

### 2.6. Evaluation of the LFIA for Direct Detection in Positive Blood Culture Bottles from the Hospital

We performed a preliminary testing with four spiked blood culture bottles (three isolates of *S. aureus* and one isolate of *S. haemolyticus*) that had been flagged as positive by a Render Automated Blood Culture System (Zhuhai Meihua Medical Technology Limited, Zhuhai, China). Furthermore, the LFIAs were initially evaluated for testing of *S. aureus* in positive blood culture bottles from the hospital. Briefly, 20 positive BacT/Alert^®^ blood culture bottles (FA Plus for aerobic; bioMérieux, Marcy l’Etoile, France) flagged as positive by the BacT/Alert^®^ Virtuo Microbial Detection System (bioMérieux, Marcy l’Etoile, France) in Srinagarind Hospital (a 1466-bed university hospital, Northeast of Thailand) were tested directly following our protocol (Figure 1). The results of the LFIA were compared with the routine methods, including biochemical tests (conventional or VITEK 2 system (bioMérieux, Marcy l’Etoile, France)) and/or commercial latex agglutination test (Oxoid Ltd., Basingstoke, Hampshire, UK) for *S. aureus*. The percent of agreement and the Cohen’s Kappa index value were calculated using the free software vassarStats (http://vassarstats.net/). The Kappa index value was interpreted as follows: ≤0, poor agreement; 0.01 to 0.20, slight agreement; 0.21 to 0.40, fair agreement; 0.41 to 0.60, moderate agreement; 0.61 to 0.80, substantial agreement; 0.81 to 1, almost perfect or perfect agreement [24,25].

All tests were performed in duplicate. If the second result was in disagreement with the first result, the isolate was tested a third time and the modal result was used for the final result. The visual inspection was performed blindly by three independent observers.

### 2.7. LFIA Imaging and Quantitative Analysis of Signal

The LFIA was imaged with a smartphone (Huawei Nova 2i, Huawei Base, Shenzhen, China) inbuilt camera at a fixed setting (ISO 50). The smartphone camera was placed at a 90° angle and a distance of 10 cm above the LFIA. The intensity of the test lines was analyzed using ImageJ software (version 1.53a) (National Institute of Health, Bethesda, MD, USA). The intensity of the LFIA signal (test line) was expressed as peak area [26].

### 2.8. Stability of the LFIA Strip

To evaluate the stability of the developed LFIA, several LFIA strips were stored at 4 °C for six months. The strips were retested monthly to check their specificity and sensitivity.

## 3. Results

### 3.1. Gold Nanoparticle Preparation and Optimization of the LFIA

The size of the gold nanoparticles was approximately 13 nm with a maximum absorption wavelength at 521 nm and the solution was wine-red in color. A minimum of 180 µg/mL of anti-protein A pAb was required to stabilize the colloidal gold nanoparticle. UV–Vis spectral absorbance showed that the peaks of gold nanoparticles shifted from 521 to 523 nm after conjugation with the antibody (Figure 2).

To optimize the LFIA, we tested the important parameters to increase the sensitivity and specificity. These included the types of sample pad, NCM types and the running buffers. UniSart^®^ CN140 produced slightly higher signal intensities than UniSart^®^ CN95. In the case of samples from blood culture, the Cytosep 1660 consistently performed better than the Milipore C048 sample pads. The highest intensity of control and test lines were obtained with buffer 4 (containing 5 mM phosphate buffer with a pH of 7.4, and 1% Triton X-100). In the optimized LFIA, the UniSart^®^ CN140 NCM, Cytosep 1660 sample pad and buffer 4 were used (Figure 3).

### 3.2. Detection Limit of the LFIA Strip

The limit of the LFIA for detection of *S. aureus* isolates in sterile culture fluid from pre-incubated aerobic culture bottles was 10^7^ CFU/mL (10^6^ CFU/reaction). Testing with the purified protein A provided a detection limit of 0.738 nM. With 72.52–595.24 nM of purified protein A, the LFIA showed results due to the “hook” effect (Figure 4).

### 3.3. Detection of S. aureus by LFIA Using Bacterial Colonies and Spiked Blood Culture Samples

The LFIA strip yielded a positive (or weakly positive) result for 55 and 51 of the 58 *S. aureus* bacterial colonies and spiked blood culture samples, respectively (sensitivity of 94.8% and 87.9%, respectively). Using the PCR as the reference method, the specificity was 100% (32/32) and 93.8% (30/32) for detection in colonies and spiked blood culture samples, respectively (Table 1). The positive and negative results obtained with an *S. aureus* and a non-*S. aureus* strain are shown in Figure 5. The control and test lines were noticeable within 30 s to 15 min. Reading of the result at 15 min is appropriate. Two *S. saprophyticus* isolates gave false positive results with very faint bands. The intensity results of all isolates (spiked blood culture samples) expressed as peak area are shown in Figure 6. The threshold value to determine a positive/negative result is 100 of peak area (from ImageJ). The peak area of ≤100 indicated a negative result, whereas that of >100 indicated a positive result.

### 3.4. Evaluation of the LFIA for Direct Detection in Positive Blood Culture Bottles from Hospital

The LFIA results of preliminary tests with spiked blood culture bottles showed perfect agreement (100% agreement with a Cohen’s Kappa index value of 1.0) with the PCR and biochemical methods. Further evaluation in 20 positive blood culture bottles from patients of Srinagarind Hospital gave 95% agreement compared with the routine methods; with a Cohen’s Kappa index value of 0.894 (almost perfect agreement). One *S. aureus* was not detected by the LFIA test (Table 2). 

### 3.5. Stability of the LFIA Strip

After storage for six months, the developed LFIA strips still gave clearly positive results with 10^7^ CFU/mL of the *S. aureus* (NCTC 10442) control strains without reduced specificity. These results indicated that the LFIA strips can be stored at 4 °C for at least six months without losing their efficacy ((signal intensity still showed higher than the positive threshold value (>100 of peak area)) (Figure 7). 

## 4. Discussion

The prevalence of bacteremia caused by *S. aureus* is up to 20% [27]. This high prevalence results from its ability to adapt to the host environment during infection. Rapid detection and identification of *S. aureus* bacteremia facilitates a prompt and adequate antibiotic therapy. Protein A is a potent virulence factor and can be taken as indicating the presence of *S. aureus* in clinical specimens [28,29]. Most *S. aureus* strains from clinical sources contained protein A associated with the cell wall or released extracellularly [30].

The LFIA, using anti-protein A antibody, has been used to detect *S. aureus* in foods [18]. However, detection of *S. aureus* in blood culture using anti-protein A antibody has not yet been reported. This may be due to the protein A secreted by *S. aureus* naturally combining with IgG molecules in blood samples, causing false negative results. Varley et al. (2016) [31] reported that IgG-protein A complexes in serum could be broken by boiling for 10 min (the protein A molecule can tolerate boiling for 10 min). On boiling, the existing bonds are likely to be broken, freeing up protein A binding sites [31]. This hypothesis supported the findings of Nilsson et al. (1990) [32]. Moreover, previous studies have shown that arginine, glycine and citrate can help to elute antibodies from a Protein A affinity column [33,34]. Therefore, we developed the LFIA strip for detection of protein A antigen in *S. aureus* extracted using a simple boiling method and diluted with a solution containing arginine, glycine and citrate. The highly positively charged arginine facilitated the dissociation of protein A and antibody complex and inhibited the aggregation of antibodies, while the negatively charged citrate inhibited the affinity of protein A and antibodies, and glycine buffered the acidic range [35].

In this study, there were some false positive and false negative samples. The false positive results were observed from two *Staphylococcus saprophyticus* isolates. This might have occurred from the effect of red blood cells that retarded the flow rate of samples, together with the production of surface protein which might have been similar to protein A, leading to falsely combining with the anti-protein A antibody. Chang and Huang reported in 1994 that some strains of *Staphylococcus* species other than *S. aureus*, including *S. capilis* subsp. capitis and *S. lentus*, could produce a small amount of protein A [36]. For the false negative results, the LFIA strip failed to detect three and seven isolates of methicillin-resistant *S. aureus* strains from colonies and positive blood culture samples, respectively. The lower number of false negative results of our LFIA strip when testing bacterial colonies may have been due to the higher inoculum size applied (>10^9^ CFU/mL) compared with the smaller number of bacterial cells from spiked blood culture samples (~10^8^ to 10^9^ CFU/mL). However, interfering substances e.g., anticoagulants or blood components may have had non-specific binding with gold nanoparticle–conjugate antibody and blocked the analyte-specific binding sites of the antibody, contributing limited interaction between the LFIA and the target antigen in the sample, which might have caused a false negative result and affected the efficacy of the immunoassays [37,38]. Moreover, in the case of a very high antigen concentration (a limited number of antibodies faced with a very large number of antigen molecules), the antigen–antibody reaction may have shown less signal, which can lead to false negative or weakly positive results, the so-called “hook” effect or “prozone” effect (as shown in Figure 4). To overcome this problem, the suspected sample should be adequately diluted before testing the LFIA or use real-time reaction kinetics for the detection [39,40]. In addition, false negative results may have been due to the small amount of protein A produced by some individual *S. aureus* strains. It was found that the amount of protein A from different *S. aureus* strains secreted into staphylococcal selective broth varied substantially (85-fold differences) [36]. The extension of incubation time from 24 h to 48 h did not increase the protein A production in most strains [36]. Thus, the LFIA strip should be further developed to detect a lower number of *S. aureus* cells (<10^7^ CFU/mL). However, positive blood cultures usually contain at least 10^7^ to 10^8^ CFU/mL [41]. Wiriyachaiporn et al. [41] reported a similar detection limit of 10^6^ CFU/mL of *S. aureus* using an LFIA strip to test a non-blood sample.

To improve the sensitivity of the LFIA, we compared several types of sample pad, NCM types and running buffers. The UniSart^®^ CN140 gave slightly higher signal intensities than UniSart^®^ CN95, providing results similar to the study of Tran et al. [42]. The flow rate, pore size and protein binding capacity of NCMs have a direct effect on the sensitivity and running time of an LFIA strip. Normally, a low flow rate helps to facilitate the formation of antigen–antibody complexes at the test and control lines [42,43]. Li et al. [44] compared different types of NCMs, including Millipore Hi-Flow Plus (HF120 and HF135) and Satorius UniSart^®^ (CN95, CN140 and CN150). They found that there was no significant difference between most of the membranes, except the HF135, which consistently performed better than the CN95.

Several strategies have been described to increase the sensitivity of LFIA strips. For example, Anfossi et al. [45] reported that the LFIA for ochratoxin A detection using a silver system was 10-fold more sensitive than the conventional gold-based LFIA that otherwise used the same immuno-reagents. However, the silver reagent was relatively unstable and highly light sensitive [16]. Recently, Tsai et al. (2018) [16] reported the use of a stacking pad (additional membrane between the conjugation pad and test pad) for enhancing the sensitivity of the LFIA for detection of protein A and C-reactive protein. The stacking pad helped to extend the binding interaction of antigen and antibody. In this study, we used an arginine-glycine-citrate buffer to enhance the sensitivity of the LFIA for detection of protein A in positive blood culture samples.

Moreover, Saisin et al. [46] reported that the sensitivity and the detection limit of the LFIA can be enhanced when using the quantitative readers at the optimal camera settings [46]. A quantitative analysis of the results of the LFIA has several advantages: (i) no subjectivity in their assessments due to digital images of a test serves as proof of diagnostic decisions, reducing error caused by varying human visual abilities; and (ii) a quantitative parameter, such as the intensity of staining of the test line, can provide information related to the concentration of a target [46,47]. A simple and compact device for the quantitative readout of the LFIA should be developed further. 

Use of in-house LFIAs for detection of *S. aureus* in various biological samples, such as bacterial colonies, food and respiratory samples, have been reported (100% sensitivity and 94.7% to 100% specificity in bacterial colonies) [19,20,41]. However, an in-house LFIA for detection of *S. aureus* from positive blood culture samples has not yet been reported. Recently, BinaxNOW *Staphylococcus aureus* test (Alere Scarborough, Inc., Scarborough, ME, USA) for detection of *S. aureus* from positive blood culture bottles was evaluated [48,49,50]. Despite its high sensitivity and specificity, this commercial test is costly (approximately $10.00 per test) [48]. Although latex and coagulase tests are reliable and easy to perform in routine laboratories, the latex test requires a pure culture (at least one day) and the coagulase test takes several hours for incubation and needs intermittent observation of the plasma clotting [19]. In 1995, Chang and Huang [51] found that protein A was a better marker than coagulase for the identification of *S. aureus*. The addition of antibody to the other target antigen such as sortaseA (the enzyme first identified in the human pathogen *S. aureus*) might increase the sensitivity and specificity of the LFIA [52]. The LFIA method has several advantages, including the lack of complicated sample preparation before testing, extreme ease of use, good performance, rapid results, low cost (approximately $1.00 per test: approximately $0.03 for nitrocellulose membrane, $0.15 for sample pad, $0.08 for absorbent pad, $0.08 for conjugate pad, $0.05 for backing card, $0.41 for anti-protein A antibody (to prepare test line and conjugate), $0.04 for goat anti-chicken antibody (to prepare control line) and $0.2 for cassette), long shelf life and ease of storage. In addition, no skilled technicians or special equipment are required. Therefore, the LFIA may be suitable for *S. aureus* detection.

However, the performance of the developed LFIA has not yet been fully validated in multiple hospitals. We have initially validated the test using 20 samples from one hospital (small sample sizes of *S. aureus* were obtained due to the prevalence). The results showed high agreement (95%) with the routine diagnostic methods. Therefore, the LFIA may be an appropriate method for testing of *S. aureus* in positive blood culture samples in hospitals. However, further evaluation using larger samples is needed.

In conclusion, the LFIA exhibits high sensitivity and specificity for detection of *S. aureus* in positive blood culture samples. It could be used as an alternative method for rapid identification of *S. aureus* and has potential for routine detection, particularly in low-resource settings. Rapid results contribute to an early diagnosis, facilitate effective treatment and retard the disease progression.

## Figures and Tables

**Figure 1 diagnostics-10-00794-f001:**
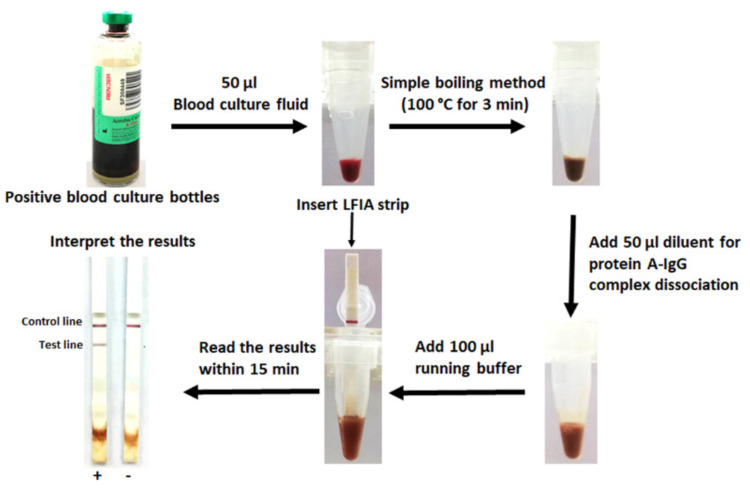
Lateral flow immunoassay protocol for direct detection of *Staphylococcus*
*aureus* in positive blood culture bottles. +, positive result; -, negative result.

**Figure 2 diagnostics-10-00794-f002:**
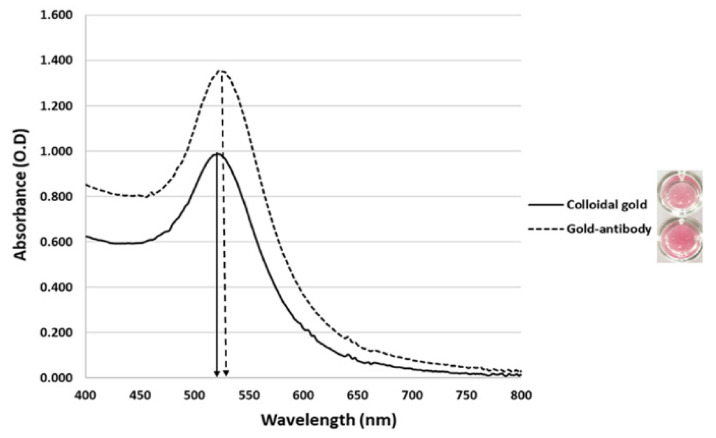
UV–Vis spectra of colloidal gold alone (solid line) and gold–antibody conjugate (dashed line).

**Figure 3 diagnostics-10-00794-f003:**
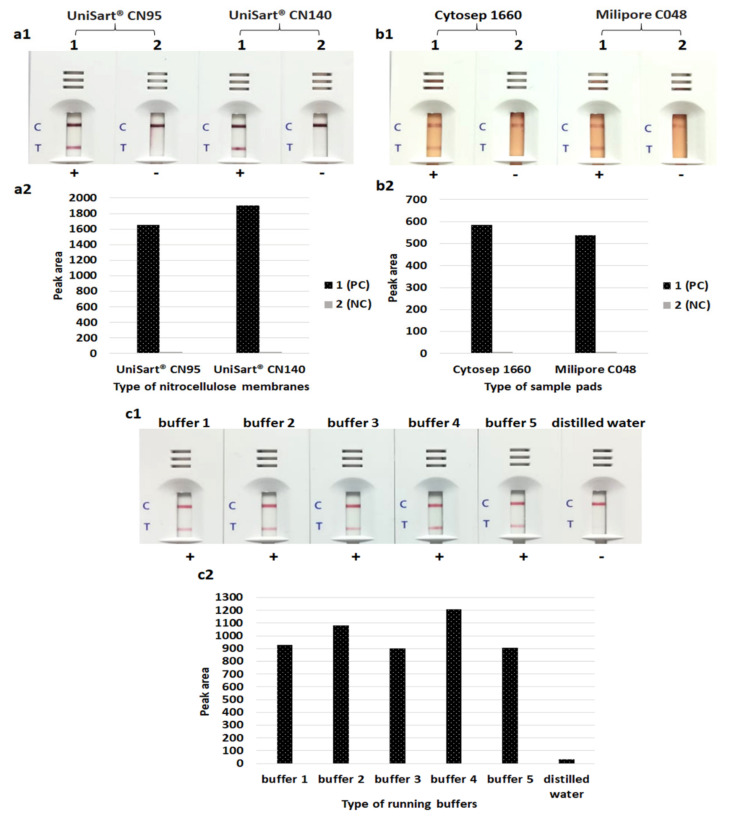
Testing of various nitrocellulose membranes (**a1** & **a2**), sample pads (**b1** & **b2**) and running buffers (**c1** & **c2**) for lateral flow immunoassay. (**a2**–**c2**) Histogram representing peak area of the test line responding from (**a1**–**c1**). 1, *Staphylococcus aureus* (NCTC10442); 2, *Staphylococcus haemolyticus* (CNSP40). PC, positive control; NC, negative control. Buffer 1, 50 mM Tris-HCl, 200 mM NaCl and 1% Triton X-100; buffer 2, 50 mM Tris-HCl and 1% Triton X-100; buffer 3, 50 mM Tris-HCl and 1% Tween 20; buffer 4, 5 mM phosphate buffer with a pH of 7.4 and 1% Triton X-100; buffer 5, 5 mM phosphate buffer with a pH of 7.4 and 1% Tween 20. C, control line; T, test line; +, positive result [two red lines of both test and control lines]; -, negative result [one red line of only the control line].

**Figure 4 diagnostics-10-00794-f004:**
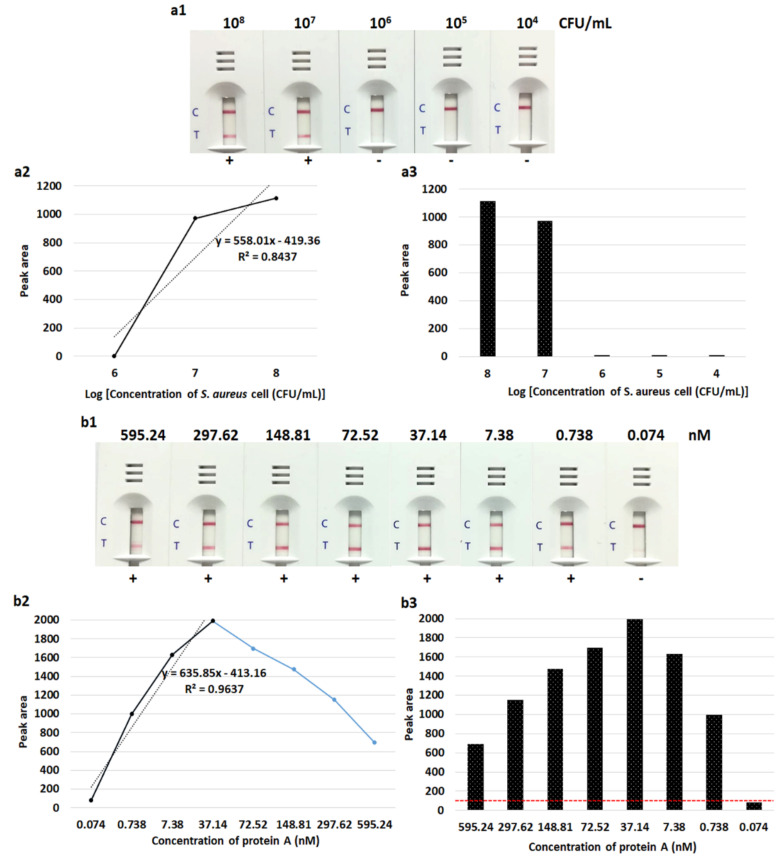
Detection limit of lateral flow immunoassay for identification of protein A in *Staphylococcus aureus* cells (**a1**–**a3**) and purified protein A (**b1**–**b3**). (**a2** & **b2**) Calibration curve of lateral flow immunoassay with different concentrations of *S. aureus* and purified protein A responding from (**a1** & **b1**). (**a3** & **b3**) Histogram representing peak area of the test line responding from (**a1** & **b1**). The red dashed line corresponds to the threshold value (peak area = 100). With 72.52–595.24 nM of purified protein A, the lateral flow immunoassay showed results due to the “hook” effect. C, control line; T, test line; +, positive result [two red lines of both test and control lines]; -, negative result [one red line of only the control line].

**Figure 5 diagnostics-10-00794-f005:**
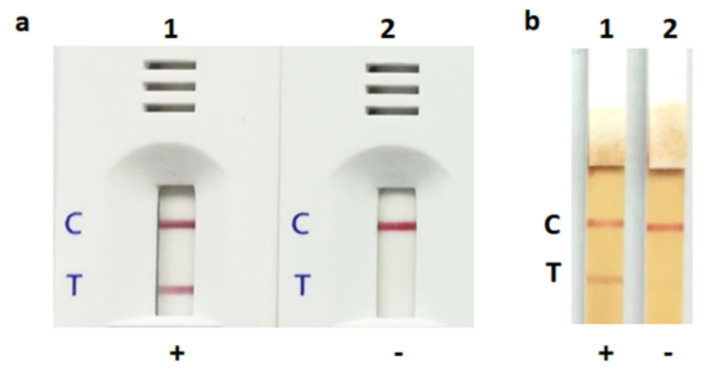
Positive and negative results of lateral flow immunoassay for detecting *Staphylococcus aureus* in bacterial colonies (**a**) and spiked blood cultures (**b**). 1, *Staphylococcus aureus* (NCTC10442); 2, *Staphylococcus haemolyticus* (CNSP40). C, control line; T, test line; +, positive result [two red lines of both test and control lines]; -, negative result [one red line of only the control line].

**Figure 6 diagnostics-10-00794-f006:**
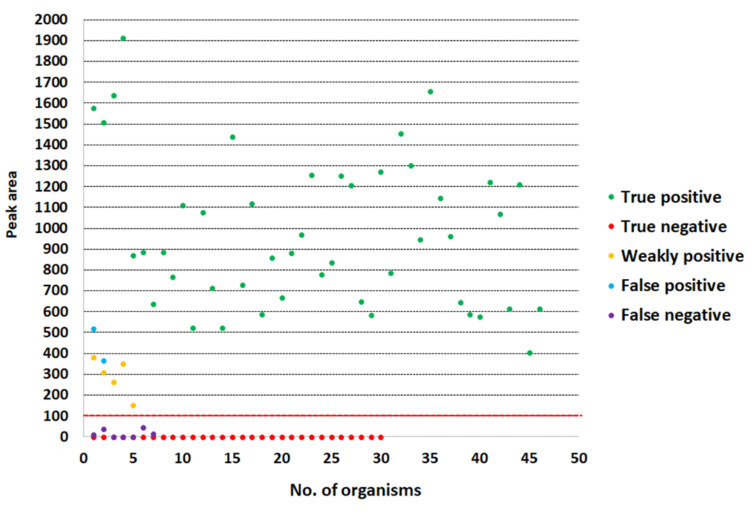
Graph plotting of all the results by groups (testing with spiked blood culture samples). The threshold value to determine a positive/negative result is 100 of peak area (red dashed line). The peak area of ≤100 indicated a negative result, whereas that of >100 indicated a positive result.

**Figure 7 diagnostics-10-00794-f007:**
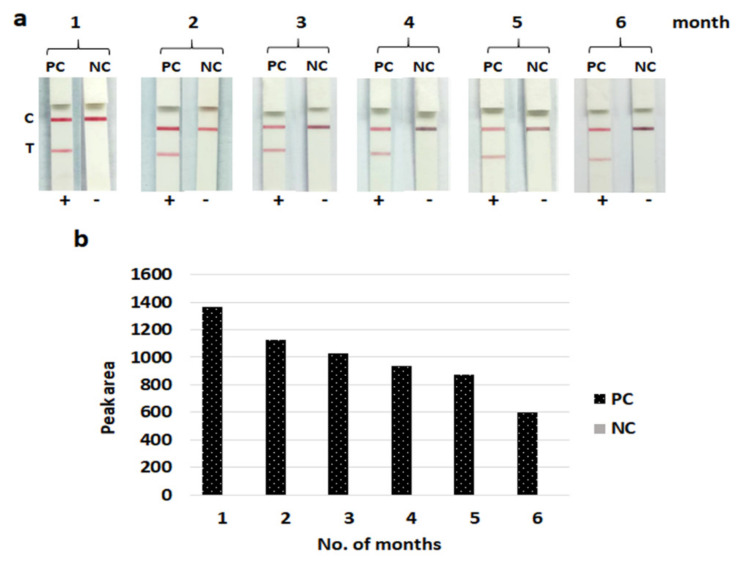
Stability of the lateral flow immunoassay strip after storage for six months (1–6). (**a**) The photo images of the lateral flow immunoassay strip. (**b**) Histogram representing peak area of the test line responding from (**a**). PC, positive control (10^7^ CFU/mL of *Staphylococcus aureus* (NCTC10442)); NC, negative control (10^8^ CFU/mL of *Staphylococcus haemolyticus* (CNSP40)). C, control line; T, test line; +, positive result [two red lines of both test and control lines]; -, negative result [one red line of only the control line].

**Table 1 diagnostics-10-00794-t001:** Sensitivity and specificity of the developed lateral flow immunoassay (LFIA) for detection of *Staphylococcus aureus.*

Organisms (*n*)	No. Isolates Positive by PCR	No. Isolates Tested by LFIA in					
		Bacterial Colonies			Spiked Blood Cultures		
		Positive	Weakly Positive	Negative	Positive	Weakly Positive	Negative
Gram-positive bacteria (76)							
*S. aureus* (58)							
*S. aureus* (54)	54	47	4	3	42	5	7
*S. aureus* (NCTC10442) (1)	1	1	0	0	1	0	0
*S. aureus SCCmec* II (1)	1	1	0	0	1	0	0
*S. aureus SCCmec* III (2)	2	1	1	0	2	0	0
Coagulase-negative staphylococci (14)							
*S. sciuri* (4)	0	0	0	4	0	0	4
*S. haemolyticus* (2)	0	0	0	2	0	0	2
*S. saprophyticus* (2)	0	0	0	2	0	2	0
*S. caprae* (1)	0	0	0	1	0	0	1
*S. chromogenes* (1)	0	0	0	1	0	0	1
*S. cohnii* spp. *urealyticus* (1)	0	0	0	1	0	0	1
*S. hyicus* (1)	0	0	0	1	0	0	1
*S. vitulinus* (1)	0	0	0	1	0	0	1
*S. xylosus* (1)	0	0	0	1	0	0	1
*Aerococcus viridans* (1)	0	0	0	1	0	0	1
*Enterococcus* spp. (3)							
*E. faecalis* (2)	0	0	0	2	0	0	2
*E. faecium* (1)	0	0	0	1	0	0	1
Gram-negative bacteria (12)							
*A. baumannii* (4)	0	0	0	4	0	0	4
*P. aeruginosa* (3)	0	0	0	3	0	0	3
*Enterobacter* spp. (1)	0	0	0	1	0	0	1
*E. coli* ATCC 25922 (1)	0	0	0	1	0	0	1
*E. coli* (1)	0	0	0	1	0	0	1
*K. pneumoniae* ATCC BAA-1705 (1)	0	0	0	1	0	0	1
*K. pneumoniae* (1)	0	0	0	1	0	0	1
Yeast (2)							
*C. albicans* (1)	0	0	0	1	0	0	1
*C. tropicalis* (1)	0	0	0	1	0	0	1

**Table 2 diagnostics-10-00794-t002:** Results of 20 positive blood culture samples detected by routine methods and lateral flow immunoassay (LFIA).

Number of Samples	Routine Methods ^a^	LFIA for *S. aureus*
1	*Staphylococcus aureus*	+
2	*Staphylococcus aureus*	+
3	*Staphylococcus aureus*	+
4	*Staphylococcus aureus*	+
5	*Staphylococcus aureus*	+
6	*Staphylococcus aureus*	+
7	*Staphylococcus aureus*	+
8	*Staphylococcus aureus*	-
9	*Micrococcus* spp.	-
10	*Micrococcus* spp.	-
11	*Staphylococcus epidermidis*	-
12	*Aerococcus viridans*	-
13	*Bacillus* spp.	-
14	*Corynebacterium* spp.	-
15	*Escherichia coli*	-
16	*Klebsiella pneumoniae*	-
17	*Salmonella* spp.	-
18	*Acinetobacter baumannii*	-
19	*Pseudomonas aeruginosa*	-
20	*Cryptococcus neoformans*	-

^a^ Routine methods included conventional biochemical tests or VITEK 2 system and/or commercial latex agglutination test.

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
