# Peer review of "Development of a Prototype Lateral Flow Immunoassay for Rapid Detection of Staphylococcal Protein A in Positive Blood Culture Samples"

_diagnostics, 2020, doi:10.3390/diagnostics10100794_

Round 1

Reviewer 1 Report

The authors describe a LFIA-based assay for detection of S. aureus based on the reactivity with protein A. The test can be applied to blood culture samples and importantly, its performance was evaluated with samples tested positive with routine detection methods. Importantly, it can be used without specialized apparatus and is stable upon storage, which makes it broadly applicable. This underlines its value, however this contribution would be more suitable as a “technical report”. The authors should expand on some experimental and measurement details, especially the number of parallels tested and visual inspection protocols (please see the comments below).

Line 87: Table 1. Please organize the headings so that the column titles are legible (in a single line)

Line 115: at 14,000xg

Line 119: microliter can be abbreviated, you use the abbreviation before in the text

Line 140: bacterial counts were performed after plating to which medium?

Line 144: Species name (S. aureus) not in italics. “PCR-based characterization” or “identification” similar would be a better title, proteins are molecules as well

Line 160: microliter can be abbreviated, you use the abbreviation before in the text

Line 161: Was visual inspection performed by independent observers and how many of them?

Line 162: phosphate buffer, pH 7.4, spiked…; and consider all concentrations in molar amounts.

Line 179: the composition of normal saline?

Line 225-227: How many parallels were performed and inspected for the reactivity? Was the evaluation done with visual inspection and how many independent individuals performed the rating?

Line 226: please state the composition of buffer 4 (if this is 5 mM phosphate buffer and 1% Triton X100, please state the pH of the phosphate buffer), the same for the title of Figure 3.

Line 230: is the evaluation of the test performed by visual inspection or can other means of quantitation be used?

Line 240, Figure 4: is the detection dependent on visual inspection? Has this been performed by independent observers? Please comment. At the same time, the intensity of test bands in figure 4b indicates a “hook” effect – less signal at higher concentrations, which would be an interesting point for discussion. There is very weak signal in the test band at the lowest concentration tested in 4b, but labelled negative, please explain.

Line 247: the figure title does not correspond to article text, please reword or place the label otherwhere: the figure shows the comparison between positive and negative results, and in the text the allowed period of the result read-out is defined.

Line 264: Excellent result; does the intensity of the signals still apply? Is the time-frame for the read-out the same?

Line 274: , and throughout the text: anti-protein A antibody or protein A-reactive antibody

Line 281: please add an explanation why these chemicals have the mentioned effect on the elution.

Line 293-294: please reword: is this a false negative or a false positive result? Please also add a tentative explanation for this effect (partial denaturation, etc.)

Line 297: reference missing

Line 334: This sentence might not be complete?

Discussion: Statements on the different tests for S. aureus detection partially overlap with what ahs already been stated in the introduction section. The authors could omit these and maybe comment on other markers suitable for S. aureus detection that could be integrated in LFIA-based screening and the potential benefit of such methods to the specificity and sensitivity.

Author Response

Dear Reviewer, 

Manuscript ID: diagnostics-913027Title: Development of a prototype lateral-flow immunoassay for rapid detection of Staphylococcal protein A in positive blood-culture samples

Thank you very much for all your comments which help improve our manuscript and become more attractive. We have revised the manuscript and answered the comments as in the following (Please see the attachment).

Yours sincerely,

Aroonlug Lulitanond

Comments and Suggestions for Authors

The authors describe a LFIA-based assay for detection of S. aureus based on the reactivity with protein A. The test can be applied to blood culture samples and importantly, its performance was evaluated with samples tested positive with routine detection methods. Importantly, it can be used without specialized apparatus and is stable upon storage, which makes it broadly applicable. This underlines its value, however this contribution would be more suitable as a “technical report”. The authors should expand on some experimental and measurement details, especially the number of parallels tested and visual inspection protocols (please see the comments below).

Answer: Dear reviewer 1,

               Thank you very much for all your comments which help improve our manuscript and become more attractive. We have revised the manuscript and answered the comments as in the following.

Reviewer 1

  1. Line 87: Table 1. Please organize the headings so that the column titles are legible (in a single line)

Answer: Thank you. We have reorganized the Table and heading as in line 95.

  1. Line 115: at 14,000xg

Answer: We changed to “at 14,000xg, 4oC” as in line 122

  1. Line 119: microliter can be abbreviated, you use the abbreviation before in the text

Answer: We changed to “μL” as in line 127.

  1. Line 140: bacterial counts were performed after plating to which medium?

Answer: We changed to “bacterial counts were performed after plating to Mueller-Hinton agar” as in line 167.

  1. Line 144: Species name ( aureus) not in italics. “PCR-based characterization” or “identification” similar would be a better title, proteins are molecules as well

Answer: We changed to “PCR-based identification of S. aureus” as in line 151.

  1. Line 160: microliter can be abbreviated, you use the abbreviation before in the text

Answer: We changed to “μL” as in line 167.

  1. Line 161: Was visual inspection performed by independent observers and how many of them?

Answer: We added the sentence “All the visual inspection was performed blindly by 3 independent observers” in lines 220-221.

  1. Line 162: phosphate buffer, pH 7.4, spiked…; and consider all concentrations in molar amounts.

Answer: We put the “,” and changed the protein A concentration to molar as “purified protein A (595.24, 297.62, 148.81, 72.52, 37.14, 7.38, 0.738, 0.074 nM)” in lines 169-171, 259 and Figure 4.  

  1. Line 179: the composition of normal saline?

Answer: We changed to “0.85% NaCl solution” as in line 188.

  1. Line 225-227: How many parallels were performed and inspected for the reactivity? Was the evaluation done with visual inspection and how many independent individuals performed the rating?

Answer: We tested each variable factors in steps. Firstly, we tested for optimum type of nitrocellulose membrane, then we used the optimum membrane to test for the best formula of buffer. Then we used the optimum membrane and optimum buffer to test for the optimum sample pad which was proper for blood sample. Each test was performed twice as indicated in lines 219-220.   The visual inspection was performed blindly by 3 independent observers as we added in line 221.

  1. Line 226: please state the composition of buffer 4 (if this is 5 mM phosphate buffer and 1% Triton X100, please state the pH of the phosphate buffer), the same for the title of Figure 3.

Answer: We added “pH 7.4” as in lines 244 and 254 (the legend of Figure 3).

  1. Line 230: is the evaluation of the test performed by visual inspection or can other means of quantitation be used?

Answer: Yes, the inspection for the results can be made by using optical equipment but in this study we aimed to develop a simple test without the need of sophisticated equipment. In the revised manuscript, we have added quantitative analysis using ImageJ program as in lines 222-226, 276-280 and Figure 3, 4, 6 and 7.

  1. Line 240, Figure 4: is the detection dependent on visual inspection? Has this been performed by independent observers? Please comment. At the same time, the intensity of test bands in figure 4b indicates a “hook” effect – less signal at higher concentrations, which would be an interesting point for discussion. There is very weak signal in the test band at the lowest concentration tested in 4b, but labelled negative, please explain.

Answer: Yes, we performed by visual inspection from 3 independent observers (line 221).

We added these sentences in the discussion section as “Moreover, in case of a very high antigen concentration (a limited number of antibodies faced with a very large number of antigen molecule), the antigen-antibody reaction may show less signal which can lead to false-negative or weakly positive results, the so-called “hook” effect or “prozone” effect (as shown in Figure 4). To overcome this problem, the suspected sample should be adequately diluted before testing LFIA or use real-time reaction kinetics for the detection” as in lines 350-355.

There is very weak signal in the test band at the lowest concentration tested in Figure 4 (b1), but labelled negative. This is because it showed very faint band and lower intensity than the positive threshold value (peak area of ImageJ <100) (Figure 4(b3) and 6).

  1. Line 247: the figure title does not correspond to article text, please reword or place the label otherwhere: the figure shows the comparison between positive and negative results, and in the text the allowed period of the result read-out is defined.

Answer: We added the text that “The positive and negative results obtained with an S. aureus and a non-S. aureus strain are shown in Figure 5.” as in lines 273-274.

  1. Line 264: Excellent result; does the intensity of the signals still apply? Is the time-frame for the read-out the same?

Answer: Thank you. Yes, the intensity of the signal still applied at 15 minutes.  The time frame for reading the test from blood samples was similar to the primarily tests. In addition, we added the results related to the stability of the device in Figure 7 and described this matter in lines 301-305).

  1. Line 274: , and throughout the text: anti-protein A antibody or protein A-reactive antibody

Answer: We changed in the text to “anti-protein A antibody” as in lines 319 and 320.

  1. Line 281: please add an explanation why these chemicals have the mentioned effect on the elution.

Answer: We added the sentences in the text as “The highly positive charged-arginine facilitated the dissociation of protein A and antibody complex and inhibited the aggregation of antibodies, while the negatively-charge citrate inhibited the affinity of protein A and antibodies and glycine buffered the acidic range” as in lines 329-332.

  1. Line 293-294: please reword: is this a false negative or a false positive result? Please also add a tentative explanation for this effect (partial denaturation, etc.)

Answer: We changed the text to “In this study, there were some false-positive and false-negative results. The false-positive results were observed from two S. saprophyticus isolates. This may occur from the effect of red blood cells that retard the flow rate of samples together with the production of surface protein which may be similar to protein A leading to false combine with the anti-protein A antibody. Chang and Huang reported in 1994 that some strains of Staphylococcus species other than S. aureus including S. capilis subsp. capitis and S. lentus could produce small amount of protein A” as in lines 333-338.

               In addition, we added the sentences “However, interfering substances e.g. anticoagulants or blood components may have non-specific binding with gold nanoparticle-conjugate antibody and block the analyte-specific binding sites of the antibody, contributing limit interaction between the LFIA and the target antigen, which may cause a false-negative results and affect the efficacy of the immuno-assays” in lines 346-350.

  1. Line 297: reference missing

Answer: We added the reference number [36], as in line 358.

  1. Line 334: This sentence might not be complete?

Answer: We added the text “Therefore, the LFIA may be suitable for S. aureus detection.” as in lines 406-407.

  1. Discussion: Statements on the different tests for  aureusdetection partially overlap with what has already been stated in the introduction section. The authors could omit these and maybe comment on other markers suitable for S. aureus detection that could be integrated in LFIA-based screening and the potential benefit of such methods to the specificity and sensitivity.

Answer: We added the text “The addition of antibody to the other target antigen such as sortaseA (the enzyme first identified in the human pathogen S. aureus) might increase the sensitivity and specificity of LFIA (Schneewind & Missiakas. Microbiol Spectr. 2019, 7, 10).” as in lines 398-400.

Reviewer 2 Report

The authors report on the development and validation of a lateral flow strip targeting Protein A in Staphylococcus aureus cells. The manuscript is nicely written, the experimental section includes details on materials characterization, assay optimization and real sample analysis. Stability of the device is described, as well. All in all, the following points are recommended to improve the quality of this manuscript:

-Relevant literature should be discussed in the introduction; for example, https://doi.org/10.1039/C3LC50169H; https://doi.org/10.3389/fbioe.2019.00069

- Details on Image acquisition settings should be described, including equipment, illumination conditions, distance, angle and so on (https://doi.org/10.3390/bios9030089).

- All the Lateral flow results (line intensities), including the optimization process, should be supported by quantitative measurements, which can be easily performed using Image J (https://doi.org/10.3390/bios9030089).

-Taking advantage of a quantitative analysis, calibration curves should be performed and the corresponding limit of detection should be based on these plots.

- Clinical evaluation of diagnostics is generally performed by employing n samples with abnormal results and 2n samples with normal results. However, the authors propose n samples in both cases. The authors are encouraged to discuss this.

- The threshold value to determine a false positive/negative result is not clear. The authors should include a graph plotting all the results by groups and the corresponding threshold to clarify this.

-The authors are encouraged to provide experimental evidence related to the stability of the device.

-The authors claim that this is a low cost device ($1.00 per test); hence, a detailed estimation of this cost should be included as supporting information.

Author Response

Reviewer 2

September 14, 2020 

Dear Reviewer,

Manuscript ID: diagnostics-913027

Title: Development of a prototype lateral-flow immunoassay for rapid detection of Staphylococcal protein A in positive blood-culture samples 

Thank you very much for all your comments which help improve our manuscript and become more attractive. We have revised the manuscript and answered the comments as in the following. We attached revised manuscript and used the "Track Changes" function in Microsoft Word (Please see the attachment). 

 Yours sincerely,

Aroonlug Lulitanond

Comments and Suggestions for Authors

The authors report on the development and validation of a lateral flow strip targeting Protein A in Staphylococcus aureus cells. The manuscript is nicely written, the experimental section includes details on materials characterization, assay optimization and real sample analysis. Stability of the device is described, as well. All in all, the following points are recommended to improve the quality of this manuscript:

  1. Relevant literature should be discussed in the introduction; for example, https://doi.org/10.1039/C3LC50169H; https://doi.org/10.3389/fbioe.2019.00069

Answer: We have added the sentence and citations in the introduction part as “Chemical and biological sensing are important tools for diagnostic in medical sciences. Plasmonic nanoparticles such as gold nanoparticles are generally reported to be biocompatible, which can be available for visual detection. Also, the using of the paper-based format has several advantages including easy to fabrication, optic transparency, biocompatibility, lightweight, and disposable technology. Therefore, the combination of plasmonic nanoparticles and paper-based leads to simple, single-use and cost-efficient analytical devices, which is useful to develop the point-of-care devices in further” (Marquez & Morales-Narváez. Front Bioeng Biotechnol. 2019, 7, 69; Yetisen et al. Lab Chip. 2013, 13, 2210-2251) in lines 66-72.

  1. Details on Image acquisition settings should be described, including equipment, illumination conditions, distance, angle and so on (https://doi.org/10.3390/bios9030089).

Answer: “The LFIA was imaged with smartphone (Huawei Nova 2i) inbuilt camera at a fixed setting (ISO 50). The smartphone camera was placed at a 90° angle and a distance of 10 cm above the LFIA. The intensity of the test lines was analyzed using ImageJ software (version 1.53a). The intensity of the LFIA signal (test line) was expressed as peak area (Yrad et al. Diagnostics. 2019, 9, 74)”. We already added this matter in lines 223-226.

  1. All the Lateral flow results (line intensities), including the optimization process, should be supported by quantitative measurements, which can be easily performed using Image J (https://doi.org/10.3390/bios9030089).

 Answer: We added the quantitative measurements which performed using ImageJ as your suggested in Figure 3, 4, 6 and 7, and lines 222-226.

  1. Taking advantage of a quantitative analysis, calibration curves should be performed and the corresponding limit of detection should be based on these plots.

Answer: We have repeated experiments and added calibration curves corresponding the limit of detection as in Figure 4.  We also added the sentence as “Moreover, Saisin et al. reported that the sensitivity and the detection limit of the LFIA can be enhanced, when using the quantitative readers at the optimal camera settings (Saisin et al. Sensors. 2018, 18, 4026). A quantitative analysis of the results of LFIA has several advantages: (i) no subjectivity in their assessments due to digital images of a test serve as proof of diagnostic decisions, making error reduction caused by varying human visual abilities; and (ii) a quantitative parameter, such as the intensity of staining the test line can provide information related to the concentration of a target (Saisin et al. Sensors. 2018, 18, 4026; 42.; Urusov et al. Biosensors. 2019, 9, 89). A simple and compact device for the quantitative readout of LFIA should be developed in further.” in lines 381-387.

  1. Clinical evaluation of diagnostics is generally performed by employing n samples with abnormal results and 2n samples with normal results. However, the authors propose n samples in both cases. The authors are encouraged to discuss this.

Answer: This data is a preliminary study. It had some limitations such as small sample sizes of S. aureus for clinical evaluation of diagnostics were obtained due to the prevalence of S. aureus in positive blood-culture sample during those study period, thus larger samples should be evaluated in further studies. We already described these matters in lines 408-413.

  1. The threshold value to determine a false positive/negative result is not clear. The authors should include a graph plotting all the results by groups and the corresponding threshold to clarify this.

Answer: We added the sentence as “the threshold value to determine a positive/negative result is 100 of peak area (from Image J). The peak area of ≤ 100 indicated a negative result, whereas that of > 100 indicated a positive result” in lines 276-280.  We included a graph plotting of all the results by groups (testing with spiked blood-culture samples) and the corresponding threshold to clarify this in Figure 6 (lines 285-288).

  1. The authors are encouraged to provide experimental evidence related to the stability of the device.

Answer: We added experimental evidence related to the stability of the device as “After storage for 6 months, the developed LFIA strips still gave a clearly positive results with 107 CFU/mL of the S. aureus (NCTC 10442) control strains without reduced specificity. These results indicated that the LFIA strip can be stored at 4°C for at least 6 months without losing their efficacy [signal intensity still showed higher than the positive threshold value (>100 of peak area)]” in lines 301-305 and Figure 7.

  1. The authors claim that this is a low cost device ($1.00 per test); hence, a detailed estimation of this cost should be included as supporting information.

Answer: This is a low cost device [approximately $1.00 per test: approximately $0.03 for nitrocellulose membrane, $0.15 for sample pad, $0.08 for absorbent pad, $0.08 for conjugate pad, $0.05 for backing card, $0.41 for anti-protein A antibody (for prepare test line and conjugate), $0.04 for goat anti-chicken antibody (for prepare control line) and $0.2 for cassette]. We state this matter in lines 402-407.

Round 2

Reviewer 1 Report

Authors have responded to the comments and clarified the issues raised for the first version of the manuscript. I can now recommend the manuscript for publication.

Reviewer 2 Report

The manuscript was revised according to my previous comments and suggestions. The manuscript is suggested for publication.